# Cerebral Aneurysm: Filling the Gap Between Pathophysiology and Nanocarriers

**DOI:** 10.3390/ijms252211874

**Published:** 2024-11-05

**Authors:** Corneliu Toader, Mugurel Petrinel Radoi, Christian-Adelin Covlea, Razvan-Adrian Covache-Busuioc, Milena Monica Ilie, Luca-Andrei Glavan, Antonio-Daniel Corlatescu, Horia-Petre Costin, Maria-Daria Gica, Nicolae Dobrin

**Affiliations:** 1Department of Neurosurgery, “Carol Davila” University of Medicine and Pharmacy, 050474 Bucharest, Romania; corneliu.toader@umfcd.ro (C.T.); christian-adelin.covlea0720@stud.umfcd.ro (C.-A.C.); razvan-adrian.covache-busuioc0720@stud.umfcd.ro (R.-A.C.-B.); milena-monica.ilie0720@stud.umfcd.ro (M.M.I.); luca-andrei.glavan0720@stud.umfcd.ro (L.-A.G.); antonio.corlatescu0920@stud.umfcd.ro (A.-D.C.); horia-petre.costin0720@stud.umfcd.ro (H.-P.C.); maria-daria.gica0720@stud.umfcd.ro (M.-D.G.); 2Department of Vascular Neurosurgery, National Institute of Neurology and Neurovascular Diseases, 077160 Bucharest, Romania; 3“Nicolae Oblu” Clinical Hospital, 700309 Iasi, Romania; dobrin_nicolaie@yahoo.com

**Keywords:** nanoparticles, intracranial aneurysms, pathophysiology, nanocarriers, inflammation, endothelial dysfunction, drug delivery

## Abstract

Intracranial aneurysms, characterized by abnormal dilations of cerebral arteries, pose significant health risks due to their potential to rupture, leading to subarachnoid hemorrhage with high mortality and morbidity rates. This paper aim is to explore the innovative application of nanoparticles in treating intracranial aneurysms, offering a promising avenue for enhancing current therapeutic strategies. We took into consideration the pathophysiology of cerebral aneurysms, focusing on the role of hemodynamic stress, endothelial dysfunction, and inflammation in their development and progression. By comparing cerebral aneurysms with other types, such as aortic aneurysms, we identify pathophysiological similarities and differences that could guide the adaptation of treatment approaches. The review highlights the potential of nanoparticles to improve drug delivery, targeting, and efficacy while minimizing side effects. We discuss various nanocarriers, including liposomes and polymeric nanoparticles, and their roles in overcoming biological barriers and enhancing therapeutic outcomes. Additionally, we discuss the potential of specific compounds, such as Edaravone and Tanshinone IIA, when used in conjunction with nanocarriers, to provide neuroprotective and anti-inflammatory benefits. By extrapolating insights from studies on aortic aneurysms, new research directions and therapeutic strategies for cerebral aneurysms are proposed. This interdisciplinary approach underscores the potential of nanoparticles to positively influence the management of intracranial aneurysms, paving the way for personalized treatment options that could significantly improve patient outcomes.

## 1. Introduction

Cerebral aneurysms are abnormal dilations of cerebral arteries that predominantly occur at the bifurcations of the Circle of Willis, mainly in the anterior circulation. They develop in patients with risk factors such as atherosclerosis, alcohol abuse, smoking, and hypertension. The rupture of an aneurysm has a very high mortality and morbidity rate, leading to a subarachnoid hemorrhage [1,2]. Cerebral aneurysms have a prevalence ranging from 1% to 5% in the general population, and the annual risk of rupture is approximately 0.7% for unruptured aneurysms. However, this risk increases significantly with the aneurysm size, location, and patient-specific factors such as hypertension and smoking history [3].

In recent years, current practices for managing unruptured intracranial aneurysms have increasingly favored endovascular treatment options. For a ruptured aneurysms, endovascular treatment has become the primary method of management [4].

In this article, we explore the exciting potential of using nanoparticles to treat intracranial aneurysms, a serious condition affecting blood vessels in the brain. By comparing them with other types of aneurysms, we try to find common ground and differences that could guide better treatment options. By learning from how other aneurysms are treated, we seek to find new ideas that could be applied to brain aneurysms. Our research highlights studies from outside the brain context, hoping to uncover insights that could lead to better treatments.

## 2. Physiopathology of Aneurysm

Aneurysms most frequently develop in areas where blood flow causes high wall shear stress, which is why they commonly occur at vessel bifurcations. It is believed that high wall shear stress (WSS) is involved in initiating cerebral aneurysm formation, while low WSS leads to its progressive development through induced proinflammatory conditions. Low WSS has a negative effect on endothelial cells and is an important contributor to arterial wall remodeling [5,6]. The frequency of a subarachnoid hemorrhage due to aneurysm rupture is estimated to be 10 to 15 cases per 100,000 individuals per year, with mortality rates as high as 50% [6].

The reduction in WSS can serve as a predictive parameter indicating the risk of aneurysm rupture [7]. For instance, studies have demonstrated that intracranial aneurysms located in the internal carotid artery, an area characterized by low wall shear stress, are associated with a higher likelihood of aneurysm rupture [8].

Low WSS causes changes in the endothelium, including alterations in the secretion and disorganization of its structure. One of the essential problems is the decrease in vasodilatory and antioxidant molecules, while increasing the production of vasoconstrictive molecules [9]. Moreover, low shear stress favors apoptosis, disturbed shear stress causes an increase in the endothelial turnover, and laminar shear stress induces cell cycle arrest in the G0 or G1 phases [10]. There is evidence suggesting that this endothelial dysfunction caused by hemodynamic stress leads to aneurysm formation, and cells begin to produce inflammatory molecules that will result in the recruitment of macrophages [11] (Figure 1).

This recruitment of leukocytes will lead to the amplification of inflammation through the secretion of inflammatory molecules. Matrix metalloproteinases MMP-2 and MMP-9 play an important role, with their production stimulated by IL-8 and TNF-alpha. It is worth noting that endothelial cells exposed to hemodynamic stress also produce TNF-alpha [12]. As shown in Figure 1, endothelial dysfunction is characterized by macrophage infiltration, leading to an inflammatory response that exacerbates aneurysm development.

TNF-alpha is a key proinflammatory cytokine involved in various inflammatory processes. It stimulates the production of MMP-9 in several cell types, including monocytes and macrophages. MMP-9, along with MMP-2, contributes to the degradation of the extracellular matrix and basement membrane, facilitating the migration of inflammatory cells to the site of inflammation [13,14,15].

In a study conducted by Aoki et al. [15], it was highlighted that the NF-κB pathway can be activated by TNF-alpha, as evidenced by the fact that in TNFR-1 heterozygous and TNFR1-deficient mice, the incidence of intracranial aneurysms was reduced compared to wild-type mice [14]. WSS can also induce PGE2-EP2 receptor signaling, activating NF-κB in endothelial cells, leading to chronic inflammation in the walls of the aneurysm [15].

One of the most significant roles of NF-kB is its involvement in the development of intracranial aneurysms. It does this by promoting the recruitment and activation of macrophages. NF-kB transactivates genes that are crucial for cell adhesion, such as Vascular Cell Adhesion Molecule-1 (VCAM-1), as well as chemotactic molecules like Monocyte Chemoattracting Protein-1 (MCP-1) [16]. In addition to increasing the expression of these molecules, NF-kB also regulates pro-inflammatory genes, including Interleukin-1 beta (IL-1β), Cyclooxygenase-2 (COX-2), and Inducible Nitric Oxide Synthase (iNOS) [17].

NFkB is also involved in the disorganization of the wall matrix, firstly by decreasing the expression of lysyl oxidase and procollagens, and secondly by increasing the expression of metalloproteinases [18,19].

In addition to endothelial damage, aneurysms also involve the impairment of smooth muscle cells. Due to vascular inflammation, a phenotypic transition of vascular smooth muscle cells occurs at the level of the aneurysm. These muscle cells reduce their contractility by decreasing the number of myofilaments [19,20]. They then migrate into the hyperplastic intima, especially at the neck of the aneurysm. These modified muscle cells express CD68 and contain lipid deposits, resembling atherosclerosis [21,22].

At the neck of the aneurysm, platelet-derived growth factor (PDGF)-BB is expressed in endothelial cells where high WSS exists, and this molecule assists in the modification of smooth muscle cells in the vessel [20].

It is known that smooth muscle cell (SMC) apoptosis occurs in intracranial aneurysms [21]. This process appears to take place through cathepsin B, which activates caspase-3, mediating apoptosis via the JNK signaling pathway [22]. Additionally, TNF-alpha is an initiator of apoptosis in cerebral vascular muscle cells, and the amount of apoptotic cells correlates with a ruptured aneurysm [23].

Recent studies have demonstrated the role of infiltrated macrophages in the formation and rupture of intracranial aneurysms. Specifically, CNS-associated macrophages (CAMs) are involved in infiltrating the arterial wall. Reducing these macrophages within the aneurysm using clodronate liposomes has been shown to improve outcomes, decrease aneurysm formation, and lower the risk of rupture [24]. Additionally, the reduction in CAMs led to a decrease in neutrophils, indicating a reduction in inflammation [19,25]. However, studies present conflicting results. Some researchers argue, supported by their studies, that the use of clodronate liposomes in a rabbit model of an intracerebral aneurysm does not alleviate aneurysmal damage. Additionally, clodronate liposomes did not lead to a decrease in the levels of MMP2 and MMP9 in the aneurysm walls. Thus, from these studies, it appears that macrophages may not be the only key factor in the development of aneurysms [26].

In addition to discussing the basic mechanisms involved in the development of aneurysms, we will also examine other mechanisms in the section dedicated to nanocarriers.

## 3. Histopathology of Intracranial Aneurysms

We addressed the intracranial aneurysm at the molecular level. Now we will approach it on a microscopic level. The wall of an aneurysm undergoes significant modifications due to inflammatory processes. This degeneration intensifies as it progresses from the neck of the aneurysm towards the dome. Notably, the loss of smooth muscle cells is most pronounced at the dome level. These changes correlate with the fact that the dome of the aneurysm is the most common site of rupture. In the wall, the recruitment of fibroblasts occurs, and the smooth muscle cells become disorganized. The vessel wall thickens and becomes hypocellular, characterized by an increase in the collagen content. At the same time, organizing thrombi develop within the lumen of the vessel [27]. As the inflammation progresses, the wall becomes irregular and thin, with sparse layers of collagen and persistent hypocellularity [28,29,30].

A study by Frösen et al. [31] classified 66 aneurysm domes into 4 qualitative morphological groups: type A: an endothelialized wall with linearly aligned smooth muscle cells (rupture risk: 42%); type B: a thickened wall with disorganized smooth muscle cells (rupture risk: 55%); type C: a hypocellular wall with myointimal hyperplasia or organizing thrombosis (rupture risk: 64%); and type D: an extremely thin thrombosis-lined hypocellular wall (rupture risk: 100%). This classification made by Frösen shows us the structural characteristics of intracranial aneurysms that can lead to their rupture. This study demonstrates the correlation between aneurysm morphology and its risk of rupture, with type D having the highest risk of rupture [30]. The ruptured aneurysm wall is more often characterized by being disorganized, thinner, and hypocellular with an organized thrombus present [31].

## 4. Nanocarriers Used in Medicine

Nanocarriers are colloidal systems that facilitate the targeted delivery of therapeutic agents, providing a means to overcome many limitations associated with traditional drug delivery methods, such as poor solubility, rapid degradation, and nonspecific distribution [32]. Common types of nanocarriers include liposomes, micelles, polymeric nanoparticles, and dendrimers. These systems range in size from 1 to 1000 nm and are engineered to improve the bioavailability, stability, and therapeutic efficacy of drugs [33,34,35].

Liposomes are spherical vesicles composed of lipid bilayers, capable of encapsulating both hydrophilic and hydrophobic drugs. They offer enhanced circulation times and reduced immune clearance when surface-modified with polyethylene glycol (PEG), a process known as PEGylation. This modification enables liposomes to evade the immune system and achieve prolonged systemic circulation, which is crucial for reaching target tissues like inflamed aneurysms [36,37].

Micelles, formed from the self-assembly of amphiphilic molecules in aqueous environments, are particularly effective in solubilizing hydrophobic drugs and enhancing their bioavailability. Their small size allows for passive accumulation in inflamed or diseased tissues via the enhanced permeability and retention (EPR) effect, a mechanism often leveraged in the treatment of conditions like intracranial aneurysms [35].

Polymeric nanoparticles are solid, biodegradable particles made from materials like polylactic acid (PLA) and poly (lacticcoglycolic acid) (PLGA). These nanoparticles provide the controlled release of encapsulated drugs, ensuring a sustained therapeutic effect over time. Their surfaces can be functionalized with targeting ligands to improve the specificity of drug delivery, allowing them to bind selectively to cells or tissues associated with aneurysm pathogenesis [36].

Dendrimers are highly branched, nanoscale macromolecules with a well-defined, tree-like structure. Their surface groups can be modified with functional ligands to target specific cellular receptors, thereby increasing the precision of drug delivery to aneurysmal tissues. Dendrimers are particularly useful for delivering multiple therapeutic agents simultaneously, offering a platform for combination therapies [37].

These nanocarriers improve the pharmacokinetics and pharmacodynamics of therapeutic agents, enhancing drug delivery to difficult-to-reach areas, such as the brain, by overcoming biological barriers like the blood–brain barrier (BBB) [38]. In the context of intracranial aneurysms, nanocarriers such as polymeric nanoparticles and liposomes offer promising solutions for the delivery of drugs like Edaravone and Tanshinone IIA, which suffer from poor bioavailability and solubility. By encapsulating these drugs within nanocarriers, their stability and targeting potential can be significantly improved, leading to better therapeutic outcomes [39,40,41,42,43,44].

One of the major challenges in nanocarrier-based drug delivery is evading the body’s immune system. To circumvent immune detection, biomimetic coatings are commonly employed. Among these, PEG is the most widely used polymer due to its non-toxic and biocompatible properties [40]. PEGylation helps reduce opsonization, phagocytosis, and nanoparticle aggregation, thus prolonging the circulation time of nanocarriers in the bloodstream. However, an emerging concern is the development of anti-PEG antibodies [41]. While some individuals may develop these antibodies following the administration of PEG-containing formulations, others may possess pre-existing antibodies. Studies in rodent models have shown that PEGylated liposomes can induce the production of these antibodies, which accelerates the clearance of PEG-coated nanoparticles, thereby reducing their therapeutic efficacy [45,46,47,48]. In addition to PEG, other biomimetic materials, such as platelet-derived coatings, red blood cell membranes, and macrophage membrane coatings, have been explored to further enhance immune evasion [44]. These alternative coatings aim to improve the biocompatibility and longevity of nanocarriers in circulation by mimicking the body’s natural cellular components [45]. There are two primary targeting strategies for nanocarriers: active and passive targeting. Active targeting relies on the functionalization of nanocarrier surfaces with ligands that specifically bind to molecules overexpressed on the surface of target cells. These ligands can include peptides, small molecules, or antibodies designed to interact with receptors uniquely present or upregulated in the pathological tissue, such as inflamed aneurysms [45]. By exploiting these molecular interactions, active targeting enhances the precision of drug delivery, ensuring that therapeutic agents accumulate predominantly at the desired site [49,50]. In contrast, passive targeting leverages the EPR effect, a phenomenon commonly observed in inflammatory and tumor tissues. In these environments, increased vascular permeability allows nanoparticles to accumulate more readily in diseased tissues without the need for specific ligand–receptor interactions. This method is particularly useful in conditions like aneurysms, where localized inflammation can facilitate nanoparticle penetration. However, passive targeting lacks the specificity of active targeting and may result in off-target effects, thereby limiting its overall therapeutic efficacy [51,52,53,54,55,56,57]. Together, these strategies represent key avenues for improving the delivery of therapeutic agents in complex diseases like intracranial aneurysms, though each approach is accompanied by its own set of limitations and challenges, including immune clearance and targeting precision. Further research into optimizing nanocarrier design, including surface modifications and biomimetic coatings, is crucial for overcoming these obstacles and enhancing clinical outcomes.

## 5. Monitoring and Identifying Cerebral Aneurysms at Risk of Rupture

The effective identification and monitoring of intracranial aneurysms, particularly those at high risk of rupture, is an ongoing challenge in clinical practice. Macrophage infiltration has been shown to be strongly correlated with aneurysm rupture, suggesting that inflammatory processes are critical contributors to aneurysm instability. When comparing ruptured and unruptured aneurysms, macrophage infiltration is significantly more prevalent in the ruptured cases, making it a valuable biomarker for identifying aneurysms likely to rupture [30]. Nanoparticles have emerged as useful tools for imaging aneurysms. Ultrasmall superparamagnetic iron oxide particles (USPIOs), composed of an iron oxide core and a hydrophilic coating, have demonstrated the ability to target and accumulate in macrophages [58,59,60].

This accumulation enables the use of USPIOs as contrast agents in magnetic resonance imaging (MRI), facilitating the visualization of inflammatory activity within aneurysms. Among these particles, ferumoxytol, initially developed for the treatment of anemia in chronic kidney disease patients, has proven particularly effective for this purpose [58,61]. Ferumoxytol accumulates in macrophage-rich regions of aneurysm walls, and its presence within 24 h of administration has been linked to aneurysms that are at high risk of rupture within the following six months [59,62]. As a diagnostic tool, ferumoxytol has an excellent safety profile, with no significant adverse events reported in its approved use for anemia [50]. However, when used repeatedly in aneurysm imaging, concerns regarding iron overload arise. Excessive iron deposition may lead to oxidative stress and damage to surrounding tissues, requiring further investigation into the long-term safety of repeated ferumoxytol use in this context [51].

Beyond iron oxide nanoparticles, other nanomaterials have shown potential in aneurysm imaging. Immunoliposomes, which are liposomes functionalized with specific antibodies, provide another method for identifying high-risk aneurysms. By targeting α-smooth muscle actin (αSMA) on vascular smooth muscle cells (VSMCs) and CD31 on endothelial cells, these liposomes offer a dual-targeting approach. The loss or dysfunction of VSMCs is a significant factor in aneurysm wall weakening, while endothelial dysfunction is closely associated with an increased rupture risk [52]. Through these specific targeting mechanisms, immunoliposomes can identify regions undergoing both structural remodeling and endothelial degradation, which are strong indicators of aneurysm instability [53]. Additionally, these liposomes can be loaded with contrast agents such as gadolinium, further enhancing their visibility in MRI scans and providing a comprehensive imaging modality that combines both inflammatory and structural markers. Despite the promising advancements in the use of USPIOs, immunoliposomes, and other nanomaterials for aneurysm imaging, several limitations and risks must be acknowledged [53]. Iron oxide nanoparticles, such as ferumoxytol, raise concerns about long-term iron accumulation in tissues. While ferumoxytol has been deemed safe for anemia treatment, the potential for iron overload through repeated imaging applications remains a significant issue. Excessive iron in the body can catalyze the formation of reactive oxygen species (ROS), leading to oxidative damage and inflammation, particularly in patients with impaired iron clearance [54].

In addition, gold nanoparticles (AuNPs), which have been explored for use in computed tomography (CT) imaging, present their own set of challenges. Although these nanoparticles provide excellent contrast due to their high electron density, their tendency to accumulate in non-target tissues, such as the liver and spleen, raises questions about their long-term safety and biocompatibility. The slow clearance of gold nanoparticles from the body could result in chronic exposure and potential toxicity, limiting their widespread use in clinical applications [55].

Similarly, quantum dots (QDs), renowned for their fluorescence and use in optical imaging, are associated with risks due to their heavy metal content, particularly cadmium [56]. The release of toxic components from QDs, especially in the harsh oxidative environment of aneurysm walls, could lead to significant cytotoxicity. The development of biodegradable alternatives is necessary to mitigate these risks while maintaining the advantages offered by QDs in terms of imaging sensitivity and resolution [57].

Addressing these challenges requires the development of biodegradable nanomaterials that can be safely metabolized and cleared from the body after their diagnostic role is fulfilled [58]. Polymeric nanoparticles, which degrade into non-toxic byproducts, offer a promising alternative to metal-based nanoparticles. Furthermore, the combination of theranostic nanocarriers, which integrate both therapeutic and diagnostic functions, could provide real-time feedback on the aneurysm status while simultaneously delivering treatments to reduce rupture risk [59]. This dual functionality would allow for the more precise and efficient management of aneurysms, potentially improving patient outcomes [60].

The use of nanomaterials such as USPIOs, gold nanoparticles, and immunoliposomes in aneurysm imaging has demonstrated significant potential in identifying aneurysms at a high risk of rupture [62,63]. However, their long-term safety profiles, including risks of toxicity and biocompatibility, must be thoroughly evaluated before these technologies can be widely adopted in clinical practice. Continued research into biodegradable and multifunctional nanomaterials will be essential in advancing the field and improving the safety and efficacy of nanoparticle-based diagnostics.

In the context of intracranial aneurysm imaging and monitoring, various nanomaterials have been developed and studied for their unique capabilities in improving the diagnostic accuracy and therapeutic efficacy [64,65,66,67]. These nanomaterials, including USPIOs, gold nanoparticles, quantum dots, immunoliposomes, biodegradable nanoparticles, and theranostic nanocarriers, each offer distinct advantages in terms of imaging modality, sensitivity, and specificity [68]. However, they also present challenges related to safety, long-term biocompatibility, and potential toxicity [69]. The table below provides a detailed comparison of these nanomaterials, summarizing their imaging modalities, benefits, limitations, and associated risks. This comparison (Table 1) helps elucidate the current state of nanotechnology in aneurysm diagnostics and offers insights into future directions for safer, more effective nanoparticle-based imaging systems.

## 6. Potential of Using Edaravone and Tanshinone IIA with Nanocarriers for the Treatment of Intracranial Aneurysms

Edaravone is a neuroprotective agent that acts as a free radical scavenger and inhibits lipid peroxidation, a key mechanism in limiting oxidative stress. It is currently used in the treatment of patients with acute cerebral infarction and amyotrophic lateral sclerosis [70,71,72,73,74,75]. Studies have demonstrated the potential of Edaravone in treating aneurysms by reducing their size and inhibiting the degradation of the aneurysm wall [76,77]. However, despite these promising effects, Edaravone’s oral bioavailability is significantly low, limiting its clinical utility in long-term aneurysm management [78,79,80]. Aoki et al. [76] elucidated the role of ROS in the pathogenesis of cerebral aneurysms. Their study showed that while ROS-producing genes were up-regulated in aneurysmal walls, ROS-eliminating genes were suppressed, exacerbating oxidative damage. They further demonstrated that Edaravone was effective in inhibiting aneurysm formation by reducing oxidative stress and inflammation. The deletion of the p47phox gene, a gene associated with ROS production, led to reduced inflammation and aneurysm formation in murine models [63]. Despite these findings, the low oral bioavailability of Edaravone presents a major obstacle to its widespread clinical use [81,82].

To address these limitations, researchers have explored nanoparticle encapsulation as a method to enhance Edaravone’s pharmacokinetics [65]. Parikh et al. [80] developed an Edaravone-loaded lipid-based nanosystem, which significantly improved the drug’s bioavailability and enhanced its neuroprotective effects [66]. By encapsulating Edaravone within liposomes or solid lipid nanoparticles, the drug is protected from rapid degradation and clearance, allowing for a sustained release at the target site. This method also improves the drug’s ability to cross the BBB, a critical factor in treating cerebral aneurysms [68]. A study comparing standard Edaravone treatment with nanoparticle-loaded Edaravone in patients with postoperative cerebral hemorrhage found that those receiving the nanoparticle formulation showed better outcomes, including reduced cerebral edema [68]. These findings strongly suggest that the nanoparticle encapsulation of Edaravone could significantly enhance its therapeutic potential in the treatment of cerebral aneurysms, making it a prime candidate for further research.

Tanshinone IIA, a phenanthrenequinone derived from *Salvia miltiorrhiza*, has been widely used in traditional Chinese medicine for treating cardiovascular diseases and stroke [83,84,85,86,87]. It has demonstrated neuroprotective, anti-inflammatory, and antioxidant effects, making it a potential therapeutic agent for intracranial aneurysms [88,89,90]. A study by Jun Ma et al. [87] suggested that Tanshinone IIA influences the progression of aneurysms by inhibiting the NF-kB signaling pathway, thereby reducing macrophage infiltration and the expression of matrix metalloproteinases (MMP-2 and MMP-9). This inhibitory effect ultimately prevents the degradation of the aneurysmal wall and reduces the likelihood of rupture [91]. However, like Edaravone, Tanshinone IIA suffers from low aqueous solubility and poor bioavailability, which limits its therapeutic efficacy [92]. Recent advances in nanotechnology have provided a solution to these pharmacokinetic challenges. Ashour et al. [78] addressed the low bioavailability of Tanshinone IIA by encapsulating it in lipid nanocapsules. This approach significantly improved the drug’s solubility and delivery to target tissues [93].

In particular, the encapsulation of Tanshinone IIA in polymeric nanoparticles, such as PLGA (poly-lactic-co-glycolic acid) nanoparticles, allows for a controlled drug release, maintaining a sustained therapeutic concentration over time [94]. The functionalization of these nanoparticles with targeting ligands, such as peptides or antibodies that recognize inflammatory markers in the aneurysm, enhances the specific delivery of Tanshinone IIA to inflamed or degenerating aneurysmal tissues [95,96]. By reducing macrophage activity and matrix degradation, Tanshinone IIA-loaded nanoparticles have the potential to prevent aneurysm progression more effectively than the drug administered in its free form [76].

In conclusion, the encapsulation of Edaravone and Tanshinone IIA in nanoparticle delivery systems represents a significant advancement in the treatment of intracranial aneurysms. Nanoparticles not only improve the bioavailability and stability of these drugs but also allow for their targeted delivery to the aneurysmal wall, enhancing therapeutic outcomes while minimizing off-target effects. Additionally, by integrating theranostic capabilities—combining diagnostics and therapy—nanoparticles can provide the real-time monitoring of drug delivery and therapeutic efficacy. Thus, further research into the use of nanomaterial-based delivery systems for these drugs holds great promise for the development of more effective and personalized treatments for aneurysms.

## 7. Nanocarriers Targeting Inflammation

Inflammation plays a key role in the progression and rupture of cerebral aneurysms, where endothelial dysfunction leads to the expression of various cell adhesion molecules and inflammatory mediators on the endothelial surface. A study using immunofluorescence and molecular MRI techniques identified the presence of VCAM-1 and P-selectin on the endothelial surface of cerebral aneurysms, suggesting their role in leukocyte adhesion and migration into the aneurysmal wall [97].

In addition to VCAM-1 and P-selectin, other adhesion molecules such as E-selectin, MCP-1, and ICAM-1 are also expressed at the aneurysm site, further promoting leukocyte infiltration and contributing to local inflammation and tissue remodeling [98,99].

Under pathological conditions, the endothelium of cerebral vessels, like any other vessel, modifies the expression of surface molecules in response to stress signals. In the context of cerebral aneurysms, endothelial cells increase the expression of integrins, CAMs, and various receptors, which facilitate leukocyte adhesion and migration into the vascular wall, exacerbating inflammation and vessel wall weakening [100]. Targeting these endothelial surface molecules with nanocarriers represents a promising therapeutic strategy to mitigate inflammation and slow aneurysm progression.

Nanocarriers designed to specifically target endothelial adhesion molecules can help limit leukocyte infiltration into the aneurysmal wall, potentially reducing inflammation and tissue damage. For instance, PECAM-1 is constitutively expressed on the surface of endothelial cells and plays a significant role in leukocyte transmigration. PECAM-1-targeted nanocarriers have been shown to decrease the infiltration of the vascular wall by leukocytes, which could prove beneficial in aneurysms. However, due to its constitutive expression in healthy endothelial cells, particularly in tissues such as the lungs, PECAM-targeted carriers may accumulate in non-diseased regions, leading to off-target effects [101]. This highlights the need for highly specific targeting mechanisms to avoid undesired accumulation in healthy tissues.

Similarly, nanocarriers targeting ICAM-1 can reduce leukocyte infiltration in the vascular wall, which is essential in controlling the inflammatory processes that contribute to aneurysm expansion [102]. ICAM-1-targeted nanocarriers not only inhibit leukocyte migration, but are also actively endocytosed by endothelial cells through pathways such as CAM-mediated, clathrin-mediated, or caveolar endocytosis, thereby enhancing intracellular drug delivery [103]. This endocytic mechanism allows for the delivery of therapeutic agents, such as anti-inflammatory drugs or MMP inhibitors, directly into endothelial cells, further improving the therapeutic effect.

These nanocarriers could be engineered to carry anti-inflammatory agents, such as dexamethasone or non-steroidal anti-inflammatory drugs (NSAIDs), or even gene therapy vectors, designed to silence the expression of pro-inflammatory cytokines or MMPs [104]. By targeting specific adhesion molecules involved in inflammation, such as VCAM-1, ICAM-1, and PECAM-1, nanocarriers offer a promising approach to reduce endothelial dysfunction and inflammation within the aneurysmal wall [83]. Furthermore, theranostic nanocarriers, which combine both diagnostic imaging and therapeutic capabilities, could allow for the real-time monitoring of inflammation while simultaneously delivering targeted treatment [84]. Despite their potential, the use of inflammation-targeting nanocarriers presents certain limitations. One concern is the potential for off-target effects, as some of the targeted molecules, such as PECAM-1, are also expressed on healthy endothelial cells, leading to an unintended accumulation in non-affected regions like the lungs [85]. Another challenge is ensuring that sufficient nanocarriers cross the BBB to reach intracranial aneurysms, where the targeting efficiency can be hindered by the protective barrier of the brain [82]. Surface modifications such as PEGylation or the use of specific ligands can enhance BBB permeability, but improving delivery remains a critical area for further research [86].

## 8. Extrapolations from Aortic Aneurysm

Most studies on the use of nanoparticles have focused on the treatment of aortic aneurysms, given the more extensive data available regarding their pathophysiology and progression. In this review, we seek to extrapolate findings from aortic aneurysm studies to intracranial aneurysms, comparing certain pathophysiological aspects of these two aneurysm types to explore the potential use of nanoparticles in cerebral aneurysm treatment. By drawing parallels between the two, we aim to open new research perspectives for future nanoparticle-based therapies.

Cerebral arteries differ from aortic arteries in several key aspects, particularly regarding their elastin content and vascular structure. Cerebral arteries contain less elastin compared to the aorta and lack an external elastic lamina and medial elastic fibers [105]. This makes them inherently more susceptible to the formation of saccular aneurysms, particularly at arterial bifurcations where WSS gradients are highest. Despite these vulnerabilities, cerebral arteries possess a relatively thick internal elastic lamina compared to extracranial arteries, which offers some degree of protection [88]. Furthermore, cerebral arteries have a thinner adventitia and only contain vasa vasorum in the proximal portions of large intracranial arteries. In contrast, the aorta is characterized by a robust elastic network that helps maintain its structural integrity, though the loss of elastin is a critical factor in both aortic and cerebral aneurysm progression [106].

A key study by Tanweer et al. [105] highlighted several shared pathophysiological mechanisms between aortic and intracranial aneurysms, despite their structural differences. Both aneurysm types are associated with elevated wall shear stress gradients, which contribute to endothelial damage and SMC apoptosis. The degradation of the vascular wall is further exacerbated by increased levels of ROS and the upregulation of matrix metalloproteinases (MMPs), particularly MMP-2 and MMP-9, which degrade elastin and collagen, compromising the structural integrity of the vessel wall [105]. Given these common features, it is plausible that the nanoparticle-based strategies used in the treatment of aortic aneurysms could be adapted to treat intracranial aneurysms. For instance, the use of elastin-targeting nanoparticles to deliver matrix-stabilizing agents has shown success in aortic aneurysm models. These nanoparticles, which are often PEGylated for an improved circulation time, have been designed to specifically bind to regions of elastin fragmentation and deliver therapeutic agents that inhibit MMP activity and reduce inflammation. In particular, PLGA nanoparticles loaded with batimastat, an MMP inhibitor, have demonstrated significant reductions in aneurysm progression in aortic models. This approach could potentially be translated to intracranial aneurysms, where elastin degradation plays a central role in aneurysm expansion. Several promising nanoparticle-based approaches developed for aortic aneurysms could also be effective in treating intracranial aneurysms. For example, superparamagnetic iron oxide nanoparticles (SPIONs), which have been extensively used in the imaging and treatment of aortic aneurysms, offer a dual function: they can be functionalized for targeted drug delivery and simultaneously used as contrast agents in MRI to detect regions of inflammation [107]. SPIONs have been conjugated with ligands targeting macrophage-specific markers or inflammatory cytokines, allowing for the localization of nanoparticles to areas of heightened inflammatory activity. These regions can then be treated with anti-inflammatory drugs or MMP inhibitors loaded within the nanoparticles. Another avenue is the development of hyaluronic acid-based nanoparticles for elastin regeneration. In the treatment of aortic aneurysms, these nanoparticles deliver elastin-stimulating agents to the aneurysm site, promoting the regeneration of elastin fibers and restoring structural integrity to the vessel wall [108]. Given that SMC dysfunction and elastin degradation are central to the progression of both aortic and intracranial aneurysms, hyaluronic acid nanoparticles could be adapted to stimulate elastin production in cerebral arteries, potentially slowing aneurysm growth. Additionally, biodegradable polymeric nanoparticles such as PLGA-based nanocarriers have been employed in aortic aneurysm models to provide the sustained release of anti-inflammatory agents and MMP inhibitors. These nanoparticles are designed to release their therapeutic cargo over extended periods, providing long-lasting treatment effects while minimizing the need for frequent dosing [109]. This technology is particularly relevant to chronic aneurysm management, where ongoing ECM degradation and inflammation necessitate prolonged therapeutic intervention. By using biodegradable carriers, the risk of long-term toxicity is minimized, making these nanoparticles ideal for clinical translation [110]. While the mechanical forces involved in aortic and intracranial aneurysm development differ, the underlying molecular mechanisms are strikingly similar, particularly in terms of ECM degradation, oxidative stress, and inflammatory signaling. This suggests that many of the nanotechnologies developed for aortic aneurysms, including nanoparticle-mediated drug delivery and nanoparticle-based imaging techniques, could be repurposed for use in the cerebral vasculature. However, the unique challenges posed by the BBB must be addressed when developing nanoparticle systems for intracranial aneurysms [111,112,113,114,115,116,117,118]. To this end, the PEGylation of nanoparticles or the use of surface modifications that enhance BBB permeability could significantly improve drug delivery to intracranial aneurysms. There is substantial potential for the cross-application of nanotechnology between aortic and intracranial aneurysms [119]. The shared molecular pathways of elastin degradation, SMC apoptosis, and inflammatory responses provide a strong foundation for adapting the nanoparticle-based therapies developed for aortic aneurysms to treat cerebral aneurysms. As research continues to evolve, future studies should focus on the optimization of nanoparticle formulations to improve BBB penetration, as well as the targeted delivery of therapeutic agents to aneurysmal tissues. These advancements could lead to more effective treatments for intracranial aneurysms, reducing the risk of rupture and improving patient outcomes.

## 9. Nanoparticles Targeting the Elastin

Both aortic and intracranial aneurysms share the characteristic feature of vascular wall degradation, particularly involving the loss of elastin and the apoptosis of SMCs. In intracerebral aneurysms, the degeneration of the media and loss of the internal elastic lamina are critical factors that contribute to aneurysm formation and progression [120]. Similarly, in abdominal aortic aneurysms, elastin degradation is a prominent feature that weakens the structural integrity of the vessel wall [121]. The degradation of elastin leads to the loss of elasticity and resilience in the arterial wall, making it more susceptible to aneurysmal expansion.

The degradation of elastin in aneurysms is primarily mediated by the upregulation of MMPs, particularly MMP-2 and MMP-9, which break down the ECM components, including elastin and collagen. Elastin-targeting nanoparticles have been developed to specifically address this degradation process by delivering therapeutic agents to the sites of elastin breakdown.

For example, Sinha et al. [116] developed elastin antibody (EL)-modified PEGylated poly (D,L-lactide) (PLA) nanoparticles (EL-PEG-PLA NPs), which selectively bind to the elastic laminae of the vascular wall. These nanoparticles were shown to accumulate only in regions where the elastic lamina was injured, while healthy vessels were not targeted, thereby demonstrating the targeting specificity of elastin-modified nanoparticles [116] (Figure 2).

This selective binding is critical for reducing off-target effects and ensuring that therapeutic agents are delivered precisely to the areas of elastin degradation within the aneurysm.

Furthermore, the role of MMP inhibitors is crucial in preventing the continued breakdown of elastin. Several hydroxamate-based MMP inhibitors, including batimastat, solimastat, marimastat, and prinomastat, have been investigated for their ability to inhibit MMP activity and preserve the ECM [122,123,124].

A study by Nosoudi et al. [120] demonstrated that PLA nanoparticles loaded with batimastat and conjugated with elastin antibodies effectively inhibited elastin degradation and prevented aneurysm development in a rat aortic aneurysm model [125,126]. These findings underscore the potential of elastin-targeting nanoparticles in treating aneurysms by both preventing elastin degradation and delivering MMP inhibitors to the affected site. In addition to preventing the degradation of elastin, nanoparticles can also be employed to promote elastin regeneration within the aneurysmal wall. Hyaluronic acid-based nanoparticles loaded with elastin-stimulating agents, such as tropoelastin or growth factors, have been shown to enhance the production of new elastin fibers by SMCs [127,128]. The ability to regenerate elastin is critical for restoring the mechanical properties of the vessel wall and preventing further aneurysmal dilation.

## 10. Nanoparticles Targeting Vascular Smooth Muscle Cells

SMCs play a central role in maintaining the structural integrity of the vascular wall by synthesizing and remodeling ECM components, including elastin and collagen. In aneurysmal disease, SMCs undergo apoptosis and dysfunction, leading to a compromised ECM and further aneurysm progression. Nanoparticles targeting SMCs are being developed to prevent SMC apoptosis and promote their regenerative capacity.

Nanoparticles functionalized with anti αSMA antibodies can selectively bind to SMCs in aneurysmal regions, delivering therapeutic agents such as anti-apoptotic drugs or growth factors to enhance SMC survival and function [103]. For example, PLGA nanoparticles loaded with anti-apoptotic agents have been shown to protect SMCs from apoptosis induced by oxidative stress and inflammatory signaling, thereby preserving ECM integrity and slowing aneurysm expansion [104].

A study by Carmen E. Gacchina et al. [124] demonstrated that SMCs in a rat model of aortic aneurysms could be stimulated to produce elastin when treated with hyaluronan oligomers and transforming growth factor-β1 (TGF-β1). The study also highlighted the importance of administering MMP inhibitors in conjunction with elastin-stimulating agents to prevent the degradation of the newly formed matrix [105]. This dual approach—promoting elastin synthesis while inhibiting MMP activity—is essential for effectively stabilizing the aneurysmal wall.

Building on these findings, Andrew Sylvester et al. [122] studied the use of PLGA nanoparticles loaded with hyaluronan oligomers in a rat model of induced aortic aneurysms. The nanoparticles successfully stimulated SMCs to produce elastin by increasing the activity of lysyl oxidase, an enzyme involved in cross-linking elastin fibers, thereby enhancing the stability of the regenerated elastin within the aneurysmal wall [106]. These studies demonstrate the potential of hyaluronic acid-based nanoparticles in promoting elastin regeneration and improving vessel integrity in aneurysms.

Additionally, gene therapy-loaded nanoparticles have been investigated for their potential to modulate gene expression in SMCs. By delivering siRNA or CRISPR-Cas9 systems that target pro-apoptotic genes, such as Bax or p53, these nanoparticles can prevent SMC apoptosis and promote cell survival. Similarly, nanoparticles carrying TGF-β1 gene therapy have been shown to enhance SMC differentiation and ECM synthesis, leading to the increased production of elastin and collagen within the aneurysmal wall [107]. Multifunctional nanoparticles are an innovative approach that simultaneously targets both elastin degradation and SMC dysfunction, providing a comprehensive strategy for aneurysm treatment. These nanoparticles are designed to deliver MMP inhibitors to regions of elastin degradation while also delivering growth factors or anti-apoptotic agents to SMCs, promoting tissue regeneration and maintaining vascular integrity [108].

For instance, multifunctional PLGA nanoparticles functionalized with elastin-binding peptides and αSMA-targeting antibodies have been developed to target both elastin-degrading regions and SMCs within the aneurysmal wall. These nanoparticles deliver a combination of MMP inhibitors to reduce elastin degradation and anti-apoptotic agents to protect SMCs from apoptosis. Additionally, these nanoparticles can be engineered with theranostic capabilities, incorporating SPIONs or fluorescent markers for the real-time imaging of aneurysm progression through MRI or optical imaging. This allows for both diagnosis and treatment, providing a more personalized approach to aneurysm management. Despite the significant advancements in nanoparticles targeting elastin and SMCs, several challenges remain. Ensuring targeting specificity is critical, as the off-target accumulation of nanoparticles in non-aneurysmal regions can lead to undesired side-effects. Moreover, the BBB presents a significant challenge for the treatment of intracranial aneurysms, as nanoparticles must be engineered to efficiently cross the BBB and deliver therapeutic agents to the aneurysmal site [109]. Surface modifications such as PEGylation or the use of ligand–receptor targeting (e.g., transferrin or lipoprotein receptors) have shown promise in enhancing BBB permeability, but more research is needed to optimize these strategies. The long-term safety and biocompatibility of nanoparticles must be thoroughly evaluated, particularly for non-degradable nanocarriers. The accumulation of non-degradable nanoparticles in tissues could lead to chronic inflammation or toxicity, highlighting the need for biodegradable carriers such as PLGA or hyaluronic acid-based nanoparticles. Ensuring that nanoparticles degrade into non-toxic byproducts and are safely cleared from the body is critical for minimizing adverse effects [110].

## 11. Controversies of Doxycycline

Doxycycline has been widely studied for its potential in treating aortic aneurysms due to its ability to inhibit the expression of MMPs, particularly MMP-2 and MMP-9, which are key enzymes involved in the degradation of elastin and collagen in the aneurysmal wall [111].

By reducing MMP activity, doxycycline may help stabilize the aneurysm wall and slow aneurysm progression. However, the use of doxycycline has been controversial, as clinical results have been inconsistent [112]. For example, Paghdar et al. [128] demonstrated that doxycycline could prevent the growth of abdominal aortic aneurysms (AAA) smaller than 5 cm [111]. However, other studies have shown contradictory results, with one trial indicating that doxycycline failed to significantly reduce AAA growth over a two-year period [113].

The administration of doxycycline is also associated with several side-effects, including photosensitivity, nausea, diarrhea, abdominal pain, and vomiting [114]. These side-effects limit its long-term use, particularly in older patients with aneurysms. To mitigate these issues, the use of nanoparticle-based drug delivery systems has been proposed. Targeted nanoparticles could encapsulate doxycycline, reducing its systemic side-effects while enhancing its delivery to the aneurysmal wall. For example, Sivaraman et al. [129] demonstrated the efficacy of cationic PLGA nanoparticles loaded with doxycycline, which inhibited MMP-2 activity in vitro. These nanoparticles not only improved the targeted delivery of doxycycline, but also enhanced lysyl oxidase (LOX) activity, which is essential for the cross-linking of elastin fibers and the stabilization of the ECM [115].

## 12. Nano-Sized Carriers vs. Micro-Sized Carriers

In the context of nanoparticle drug delivery, the size of the carrier is a critical factor that influences its biodistribution, accumulation, and therapeutic efficacy [129,130,131,132]. In a study by Mark Epshtein [133], the efficacy of polystyrene carriers coated with glycoprotein VI (GPVI), a platelet receptor that binds to collagen, was demonstrated in both in vitro and in vivo models [133]. These carriers accumulated preferentially in areas of the aneurysmal wall with disrupted endothelium, where collagen was exposed. The study compared nano-sized carriers with micro-sized carriers and found that nano-sized carriers did not exhibit preferable accumulation in aneurysmal sites compared to their micron-sized counterparts. This finding underscores the challenge of using nano-sized particles in aneurysm treatment, where factors such as endothelial permeability and vascular architecture may limit the accumulation of nanoparticles in the aneurysmal wall.

## 13. Conclusions

In conclusion, this review has highlighted the promising role of nanoparticles in the treatment of aneurysms, particularly in targeting elastin degradation, SMC dysfunction, and inflammation. By extrapolating findings from studies on aortic aneurysms to cerebral aneurysms, we have identified both key similarities and distinct challenges that inform the development of more targeted therapeutic strategies. This comprehensive analysis enhances our understanding of aneurysm pathology and underscores the value of interdisciplinary approaches—combining nanotechnology, vascular biology, and clinical medicine—in advancing treatment options.

Looking ahead, several critical areas of research require attention. Nanoparticle design and optimization are essential for enhancing targeting specificity, particularly in intracranial aneurysms. Developing multifunctional nanoparticles that can cross the BBB, target specific molecular pathways like MMPs or inflammatory cytokines, and offer theranostic capabilities—combining diagnostic imaging with therapy—will be key for improving outcomes in cerebral aneurysm treatment.

Another important area is improving the targeting efficiency. While nanoparticle systems show promise, challenges such as off-target accumulation and low bioavailability at the aneurysm site remain. Enhancing surface modifications, such as PEGylation, and incorporating ligand–receptor targeting could improve nanoparticle permeability in blood vessels and ensure more effective delivery to aneurysmal walls, especially where endothelial integrity is compromised.

For nanoparticles to be viable in clinical settings, clinical translation must be prioritized. While preclinical studies have demonstrated efficacy in animal models, further research is required to assess their long-term safety, biocompatibility, and therapeutic effectiveness in humans. Translational studies focusing on nanoparticle pharmacokinetics, pharmacodynamics, and potential toxicity in human patients are crucial, especially for both aortic and cerebral aneurysms.

As advancements in personalized medicine continue, genetic profiling and molecular diagnostics could be integrated with nanoparticle therapies to create customized treatment regimens. Tailoring nanoparticle formulations to individual patients’ genetic profiles could significantly improve targeting precision and therapeutic outcomes considering genetic variations that affect aneurysm progression and responses to treatment.

Future research should also explore combination therapies that utilize nanoparticles to deliver multiple therapeutic agents, such as MMP inhibitors, anti-inflammatory drugs, and gene therapies, simultaneously. These combination approaches could provide a synergistic effect by targeting multiple pathological pathways like extracellular matrix degradation, inflammation, and vascular remodeling, while also minimizing systemic side-effects.

In summary, nanoparticles hold great promise for advancing the treatment of aneurysms. However, further research is needed to optimize their design, improve the targeting efficiency, and ensure their safety and efficacy in clinical applications. By focusing on these key areas, future studies have the potential to drive significant innovations in personalized medicine and nanotechnology-based therapies, ultimately improving the quality of life for patients dealing with these challenging vascular conditions.

## Figures and Tables

**Figure 1 ijms-25-11874-f001:**
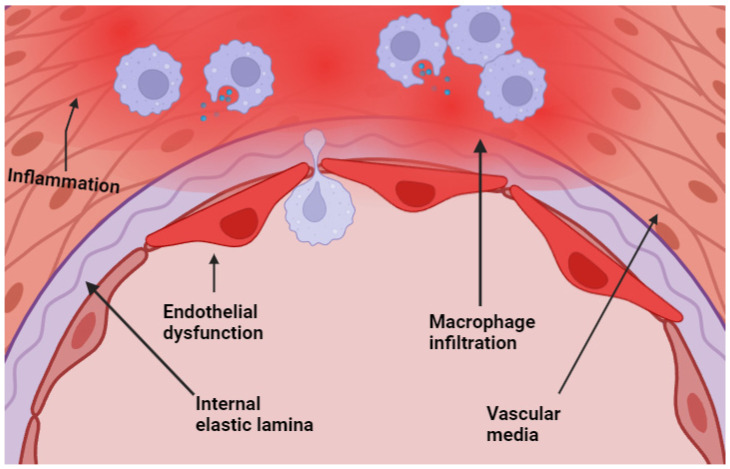
Endothelial dysfunction with macrophage infiltration at the level of the media of the cerebral arteries.

**Figure 2 ijms-25-11874-f002:**
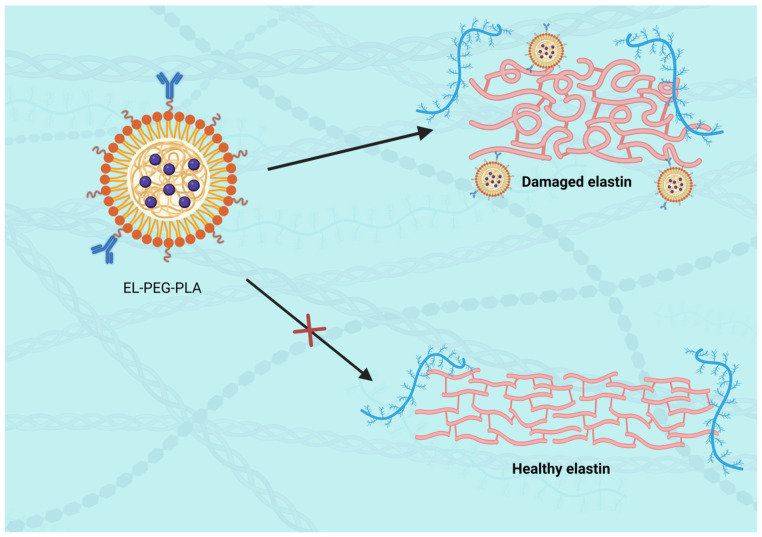
EL-PEG-PLA nanoparticles targeting elastic lamina injury.

**Table 1 ijms-25-11874-t001:** The table presents a comprehensive comparison of nanomaterials used in aneurysm imaging, detailing their respective advantages and potential risks.

Nanomaterial	Imaging Modality	Advantages	Limitations	Potential Risks
USPIOs (Ultrasmall Superparamagnetic Iron Oxide Particles)	MRI (Macrophage-targeted)	High sensitivity for macrophage activity, good contrast in MRI	Potential for iron overload, oxidative stress with repeated use	Iron overload, oxidative stress, inflammation in long-term use
Gold Nanoparticles (AuNPs)	CT (Computed Tomography)	High electron density, excellent CT contrast	Long-term accumulation in non-target tissues, potential toxicity	Accumulation in liver and spleen, long-term toxicity
Quantum Dots (QDs)	Optical Imaging (Fluorescence)	Bright fluorescence, high sensitivity for optical imaging	Heavy metal content (e.g., cadmium), risk of cytotoxicity and degradation	Cytotoxicity from heavy metal release, especially cadmium
Immunoliposomes	MRI (Dual-targeted)	Specific targeting of VSMCs and endothelial cells, high specificity in MRI	Potential immune response, clearance issues, long-term safety unknown	Possible immune clearance, potential toxicity if mis-targeted
Biodegradable Nanoparticles	MRI, CT, Optical (Multimodal)	Safe metabolism, non-toxic degradation products, applicable for multiple imaging modalities	Potentially lower sensitivity than metal-based nanoparticles, development still ongoing	Still in development, safety and long-term risks under investigation
Theranostic Nanocarriers	MRI, CT, Optical (Theranostics)	Dual function: diagnosis and therapy, real-time feedback during treatment	Complex design, need for precise targeting, risk of non-specific effects	Complexity in manufacturing, potential for off-target effects, long-term toxicity concerns

## Data Availability

Not applicable.

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
