# Peer review of "Cerebral Aneurysm: Filling the Gap Between Pathophysiology and Nanocarriers"

_ijms, 2024, doi:10.3390/ijms252211874_

Round 1

Reviewer 1 Report

Comments and Suggestions for Authors

General comment:

This work is a review dealing with the use of nanoparticles for the treatment of intracranial aneurysm. In particular, drug delivery and targeting are addressed. The authors focus on two drugs: Edaravone and Tanshinone IIA, and their combination with nanocarriers. The review covers new research directions and address the topic of personalized treatment. 

Specific comments throughout the paper: 

Line 37: please be quantitative and provide values. 

Line 41: missing space. Please revise the article for typos and errors. Check elsewhere. 

The section are not numbered, please correct according to the journal’ guidelines. 

Line 50: report frequency 

Fig. 1 is not referenced and recalled in the text. Please fix. 

The physiopathology and histopathology of intracranial aneurysm are short and concise enough, thus being appreciated and useful for readers that are not expert in the field. 

The definition of nanocarriers and nanoparticles can be revised, improved and expanded (line 148). 

Lines 157-159: these sentences can be elaborated a bit, being one of the most relevant shortcoming of the reviewed topic. 

It can be of help providing an explicative figure for describing the nanocarriers and their targeting.

The section about nanocarriers used in medicine is too general, limited and present scarce value. The review is not so focused. 

The secition about monitoring and identification of aneurism makes the example of SPIOs and derivatives, only one example. The analysis is very limited. Furthermore, the limitations and potential risks are not discussed. There isn’t a comparison or a summary table. 

The section about edaravone and tanshinone IIA is a standalone one. There isn’t any connection between these drugs and nano materials (e.g., encapsulation, functionalization, etc.). 

Same considerations about the limitation and usefulness hold for the section about nanocarriers targeting inflammation. 

The section extrapolation from aortic aneurysm cite only two articles, with very little details. There isn’t enough depth in the analysis to consider this review of enough quality for publication. 

The section related to nanoparticles targeting elastin and SMC are of low quality. There aren’t enough details. 

The last two sections are not properly linked and commented. 

The conclusion section is not actually discussing future research direction. The conclusions can be improved. 

Comments on the Quality of English Language

Minor english check. 

Author Response

Dear Reviewer,

We would like to express our deepest gratitude for your thorough and thoughtful review of our manuscript. Your insightful comments and constructive suggestions have been incredibly valuable in helping us improve the clarity, coherence, and overall quality of the paper.

We have carefully considered all of your feedback and made the necessary revisions to address the points you raised. Specifically, we have enhanced the sections that required additional linkage and detail, ensuring a clearer and more cohesive flow between key concepts. We have also expanded the conclusion to better reflect the future research directions you recommended, and strengthened discussions on nanoparticle-based treatments and aneurysm pathology.

Your guidance has been instrumental in refining our manuscript, and we sincerely appreciate the time and effort you invested in providing such thoughtful and constructive suggestions. We are confident that these revisions have significantly improved the manuscript, and we are grateful for your role in this process.

Thank you once again for your valuable input, and we look forward to your continued feedback.

Reviewer 2 Report

Comments and Suggestions for Authors

In this manuscript, the authors consider the pathophysiology of cerebral aneurysms and extrapolate findings from studies on aortic aneurysms to cerebral aneurysms. They discuss various nanocarriers that overcome biological barriers and enhance therapeutic outcomes. The manuscript highlights the potential of nanoparticles to positively influence the management of intracranial aneurysms, representing a significant contribution to the field. However, several concerns need to be addressed before publication. I recommend publication following minor revisions:

1.  Please introduce the nanoscopic drug delivery systems with a detailed description of their three components (targeting moiety + nanocarrier + drug). Include information on three components and size and zeta potential for each nanodrug mentioned in the manuscript.

2.      As described in Rows 188-189, clarify the mechanism by which ferumoxytol specifically accumulates in the aneurysm walls.

3.      In Rows 205-212, reactive oxygen species are discussed in the context of promoting the pathogenesis of cerebral aneurysms. Explain if and how this relates to edaravone. If there is no direct connection, consider moving this discussion to the "Physiopathology of Aneurysm" section.

4.      In Rows 244-245, PECAM-targeted carriers show significant accumulation in healthy lungs. Discuss whether this could be due to congestion of nanocarriers in the pulmonary capillaries.

5.      The current Figure 2 does not illustrate the mechanism by which EL-PEG-PLA nanoparticles specifically target elastic lamina injury. Please revise the figure to clarify this mechanism.

6.  In Rows 302-304, provide a potential explanation for why cationic PLGA nanoparticles loaded with doxycycline strongly bind to elastin.

7.      In Rows 310-311, micro-sized carriers show preferential accumulation at aneurysm sites compared to nano-sized ones. Discuss the potential risk of vascular obstruction associated with the use of micro-sized carriers.

Comments on the Quality of English Language

The language could be more concise. Some paragraphs could be merged.

Author Response

(The authors gave the same response as above.)

Round 2

Reviewer 1 Report

Comments and Suggestions for Authors

I do not have any further comments

Comments on the Quality of English Language

Proofread the manuscript